# Digitalized 3D Spinal Decompression and Correction Device Improved Initial Brace Corrections and Patients’ Comfort Among Adolescents with Idiopathic Scoliosis: A Single-Centre, Single-Blinded Randomized Controlled Trial

**DOI:** 10.3390/bioengineering11121246

**Published:** 2024-12-09

**Authors:** Yi Jie, Mengyao Li, Anqin Dong, Yu-Yan Luo, Chang-Liang Luo, Jing Li, Pengyuan Zheng, Xinmin Zhang, Man Sang Wong, Christina Zong-Hao Ma, Ming Zhang

**Affiliations:** 1Department of Biomedical Engineering, The Hong Kong Polytechnic University, Hong Kong SAR, China; yi620.jie@connect.polyu.hk (Y.J.); yuyan-laura.luo@connect.polyu.hk (Y.-Y.L.); chang-liang.luo@connect.polyu.hk (C.-L.L.); ming.zhang@polyu.edu.hk (M.Z.); 2Department of Rehabilitation Medical Engineering, The Fifth Affiliated Hospital of Zhengzhou University, Zhengzhou 450052, China; mengyao0930@outlook.com (M.L.); anqindong@163.com (A.D.); jingliljlj@163.com (J.L.); medp7123@126.com (P.Z.); 3Research Institute of Rehabilitation Medicine, Henan Academy of Innovations in Medical Science, Zhengzhou 450000, China; 4Zhengzhou Feilong Medical Equipment Co., Ltd., Zhengzhou 450000, China; 17737316777@163.com; 5Research Institute for Sports Science and Technology, The Hong Kong Polytechnic University, Hong Kong SAR, China; 6Research Institute for Smart Ageing, The Hong Kong Polytechnic University, Hong Kong SAR, China

**Keywords:** adolescent idiopathic scoliosis (AIS), in-brace correction, patient comfort, prognosis, vertebral number, 3D spinal decompression and correction, randomized controlled trial

## Abstract

This study aimed to evaluate the efficacy of a novel three-dimensional (3D) spinal decompression and correction device in improving the in-brace correction and patient comfort level for adolescents with idiopathic scoliosis (AIS), and to assess the impact of the number of vertebrae involved in the scoliotic curve on the correction’s effectiveness. A single-centre, single-blinded randomized controlled trial (RCT) was conducted in 110 AIS patients aged 10–18 years who were randomly allocated into four groups receiving 0–3 days of device intervention. Each session lasted for 30 min and was conducted twice daily. Significant improvements were observed in both the in-brace correction ratio and patient comfort level, particularly in the 2- and 3-day intervention groups (*p* < 0.001). The number of involved vertebrae for a scoliotic curve was positively correlated with the in-brace correction ratio in the no intervention (or 0-day) and 1-day intervention groups, while this correlation varied in the 2- and 3-day intervention groups. These findings suggested that the prolonged use of the 3D device could improve the correction ratios and patient comfort, while the role of vertebrae involvement in predicting the initial correction may require further exploration to optimize personalized treatment strategies in future clinical practice.

## 1. Introduction

Scoliosis is a common skeletal deformity affecting adolescents worldwide [1], and is characterized by the lateral curvature and rotation of the spine [2]. This condition not only affects the appearance of the affected individuals [3], but may also have long-term effects on their respiratory function and cardiac health [4,5]. Treatment options for scoliosis primarily include surgical and non-surgical approaches, with the non-surgical treatment often involving the use of spinal orthoses or braces, particularly for scoliotic cases with mild-to-moderate severity [6,7].

Brace treatment has been an effective method to prevent the progression of scoliosis and, as a result, to avoid the need for surgical intervention. Extensive research supports the positive role of brace treatments in delaying or reversing the progression of scoliosis [6,7,8,9]. However, its success rate can be influenced by various factors, including the design of the brace, the duration of wearing time by the patient, and patient adherence [10,11,12]. Nevertheless, the patient’s comfort when wearing the brace is a key factor affecting patient compliance [13]. Patient discomfort, including that caused by the excessive friction and pressure from brace and the psychological distress related to one’s self-image, can significantly impact the effectiveness of the treatment [13]. Discomfort when wearing the brace not only reduces the duration of wearing time by the patient but also directly affects the correction outcomes, ultimately deteriorating the effectiveness of the treatment [14]. It is essential to develop an approach to address the discomfort issue and improve brace treatment outcomes.

The effectiveness of in-brace correction has been considered as an important basis for predicting the final corrective outcomes for scoliosis. Van den Bogaart et al. assessed the predictive factors of brace treatment in AIS in a review, and reported that an insufficient initial correction within the brace is strongly correlated with treatment failure [15]. This indicates that the effectiveness of the initial correction is a key indicator of the success or failure of brace treatment. Research by Xu et al. demonstrated that the initial correction rate (ICR) within the brace (or in-brace correction) can serve as a predictor for the effectiveness of brace treatment, and that a higher ICR is associated with a higher probability of successful treatment outcomes [16]. All these studies support the adoption of in-brace correction to evaluate and predict the effectiveness of a brace to correct scoliotic curvature during the initial fitting in patients with AIS. In addition to in-brace correction, there might also be some additional factors contributing to the effectiveness of brace treatment outcomes, including the number of vertebrae in a scoliotic curve, which has not been studied before and merits further exploration.

To further improve brace treatment effectiveness, this study conducted a single-centre double-blinded randomized controlled trial (RCT) and explored the effectiveness of a novel 3D spinal decompression correction device in treating patients with adolescent idiopathic scoliosis (AIS). It was hypothesized that this device could enhance the in-brace correction efficiency and improve patient comfort by employing longitudinal traction, lateral three-point corrective force, and axial rotational adjustments. This study also assessed the impact of the number of vertebrae involved in a scoliotic curvature on the in-brace correction ratio and explored whether the developed treatment protocol with the device could mitigate the influence of vertebral number discrepancies on correction outcomes. This approach is expected to provide more effective non-surgical treatment options than the existing ones for patients with AIS.

## 2. Materials and Methods

### 2.1. Study Design

This single-centre, double-blinded (for outcome assessors and statisticians) RCT was conducted from May 2023 to February 2024 in the Department of Rehabilitation Medical Engineering at The Fifth Affiliated Hospital of Zhengzhou University. The research protocol was approved by the Ethics Committee of The Fifth Affiliated Hospital of Zhengzhou University (Reference number: KY2022028; Date: 23 June 2022) and followed the tenets of the Declaration of Helsinki. Signed informed consent forms were obtained from all participants and their guardians before the study. This study was retrospectively registered on 15 May 2024 in the World Health Organization (WHO) International Clinical Trials Registry Platform (ICTRP)—Chinese Clinical Trial Registry (Reference number: ChiCTR2400084383).

### 2.2. Participants

All patients were recruited at the outpatient clinic of the Department of Rehabilitation Medical Engineering at The Fifth Affiliated Hospital of Zhengzhou University. Patients diagnosed with AIS who met the eligibility criteria were recruited. Clinical diagnoses were made by a spine specialist with over 10 years of clinical experience. The inclusion criteria were as follows: 10–18 years old, diagnosed with AIS, met the criteria for brace treatment (a Cobb angle between 20 and 45 degrees), and a Risser sign grade of 0–4. The exclusion criteria were as follows: with a history of cardiovascular disease, neuromuscular disorder, psychiatric disorder, or prior conservative intervention for scoliosis.

A total of 110 patients diagnosed with scoliosis met the eligibility criteria and participated in this study. Researchers obtained the written informed consent from patients and their guardians at the outpatient clinic before data collection.

### 2.3. Randomization and Blinding

Randomization was conducted in a 1:1:1:1 ratio. The randomization sequence was prepared in advance by an independent statistician. The opaque envelopes containing the allocated intervention group numbers were also prepared by the same independent statistician. The original randomization list was kept confidential by the independent statistician and was not disclosed to the researchers. The random envelopes were opened in the presence of at least two individuals when the patients were recruited at the outpatient clinic. Due to the nature of the study, the blinding of therapists and AIS participants was not feasible. However, the study maintained blinding for the outcome assessors and data analysts to ensure their impartiality during patient evaluation and data analysis.

### 2.4. Intervention

The research team has developed an innovative 3D spinal decompression and correction device for patients with AIS. This device comprised a control platform, a longitudinal traction cable/band, a lateral three-point correction assembly, and an axial rotational adjustment mechanism located on the treatment bed. The device operated by alternately applying the corrective longitudinal traction force, lateral three-point force, and rotational force. It can be adapted to correct various types of scoliosis with different Cobb angles in patients with AIS. Specifically, the two pairs of pneumatical pressure pads (correction and counteracting/stabilizing pads) of the lateral three-point correction assembly had inflatable air bladders. The embedded force transducers in the bladders could measure and display the instantaneous lateral passive force being applied. The available longitudinal traction force generated by the traction cable/band ranged from 0 to 666 N, and the lateral corrective force generated by the three-point corrective component ranged from 0 to 500 N. The distraction impulse was 58.1 N/s, and the time error between generating longitudinal traction and lateral passive force was within 0.6 s. The traction intensity, traction mode (static or intermittent), pulling angle, and traction duration could be selected in the user interface system (Figure 1).

During the intervention, patients received a 28-min digitalized 3D scoliosis correction intervention session consisting of 7 intervention cycles, as instructed by the same certified physical therapist. Each 4-min intervention cycle included, firstly, a 2-min stretching phase with gradually increasing longitudinal traction forces and decreasing lateral corrective forces (reaching the maximum and minimum values at the end of the trial, respectively), followed by a 2-min resting phase with gradually decreasing longitudinal traction forces and increasing lateral corrective forces (reaching the minimum and maximum values at the end of the trial). The provision of longitudinal traction was derived from the treatment protocol of mechanical spinal traction for patients with low back pain [17], initiating at an intensity that did not exceed 50% of the participant’s body weight [18] and maintaining an angle that was parallel to the spine (pulling angle = 0 degree). The lateral corrective forces were manually applied to patients, with reference to each patient’s coronal radiographies and the three-point pressure system [19], by adjusting the position of the pressure pads on the apical vertebra directly or on the corresponding ribs. Reactive forces were also applied to the upper and end vertebrae and pelvis using pressure pads to prevent body shifting, following the guidelines of the three-point pressure system. The provision of lateral corrective force adhered to the criteria that equalled to 330 N or the maximum lateral corrective force that each patient could tolerate, whichever was the smaller force value.

In this study, AIS participants were randomly assigned to one of four groups. The control group received no device treatment (Group 1, no intervention or 0-day group); the first experimental group underwent 1 day of treatment with the device (Group 2, 1-day group); the second experimental group received 2 consecutive days of treatment with the device (Group 3, 2-day group); and the third experimental group received 3 consecutive days of treatment with the device (Group 4, 3-day group). The dosage of the device treatment was twice per day, with each session lasting for approximately 30 min. All device treatments were completed the day before the fitting of the orthotic braces.

All orthotic braces were designed by a senior certified orthotist with 20 years of clinical experience using computer-aided design (CAD) software (Version18.0.4, Orthotics and Prosthetics CAD, Canfit™, Vorum, Vancouver, BC, Canada). The design criterion was to achieve a correction of the patient’s in-brace Cobb angle to zero degrees, indicating a full correction, in the software simulation. The orthotic brace was finalized and adjusted by the same senior certified orthotist, who also provided guidance on its wearing to each of the AIS participants. After achieving the optimal fit, an in-brace X-ray was taken for each participant to measure the in-brace Cobb angle (Figure 2). 

### 2.5. Outcome Measures

#### 2.5.1. Primary Outcomes

This study focused on two primary outcomes:In-brace Correction Ratio: Immediately after the patients wore the spinal braces, an X-ray examination was conducted to measure the in-brace Cobb angle (Figure 3a). The in-brace correction ratio was calculated using the following equation:
In-Brace Correction Ratio = (Pre-brace Cobb Angle − In-brace Cobb Angle)/(Pre-brace Cobb Angle)(1)

This ratio could provide a quantitative measure of how effectively the brace corrected the scoliosis curvature (Figure 3a).

2.Visual Analogue Scale (VAS): After fitting the brace, the VAS was used to assess a subject’s comfort level [20]. The scale ranged from 0 to 10, with 0 meaning very comfortable and 10 meaning very uncomfortable (Figure 3b). This rating was given independently by the patient to ensure an unbiased assessment of their comfort level while wearing the brace.

#### 2.5.2. Secondary Outcomes

The number of vertebrae involved in a scoliotic curve based on the Cobb angle measurement was also analyzed. It was measured by counting the number of all the vertebrae from the upper end vertebra to the lower end vertebra of a scoliotic curve on the X-ray (Figure 3c). Whether the variations in the number of vertebrae affected the in-brace correction ratio, and whether the treatment with the device could mitigate this impact, were examined. Additionally, the correlation between the number of vertebrae and the correction ratio was analyzed.

### 2.6. Statistical Analysis

All statistical analyses of the collected data were performed using the IBM SPSS Statistics 26.0 software package (IBM Corp; Armonk, NY, USA). The normality of the variables was assessed using the Kolmogorov–Smirnov test. Analyses of variance (ANOVAs) were conducted to compare the differences in the in-brace correction ratios and comfort scores at baseline among the four groups. Post-hoc comparisons were performed using the least significant difference (LSD) method to identify the existence of specific between-group significant differences whenever a significant F-test result was obtained in the ANOVA.

Subjects were also divided into two groups based on the number of vertebrae involved in the scoliosis curvature: one group included individuals with a vertebral count below the median, and the other included those with a vertebral count above the median. The Mann–Whitney U test was then used to compare the in-brace correction ratios between these two groups. Pearson’s correlation analysis was used to explore the relationship between the number of vertebrae and the in-brace correction ratio. A two-tailed *p*-value of less than 0.05 was considered statistically significant.

## 3. Results

### 3.1. Participant Characteristics

As shown in Figure 4, between 1 May 2023 and 27 February 2024, a total of 141 patients were assessed for eligibility, of whom 116 met the inclusion criteria and did not meet the exclusion criteria. Finally, a total of 110 eligible patients participated in the study, as 6 patients refused to participate and withdrew from the study (Figure 4).

The baseline characteristics of the participants are presented in Table 1. The mean age of the entire cohort was 14.40 (standard deviation [SD] = 1.94) years. There were 83 (72%) female patients and 32 (28%) male patients, with an average age of 14.27 (SD = 1.96) years and 14.74 (SD = 2.08) years, respectively. During the trial, the per-protocol analysis included 110 patients: 30 (27%) in the control, or 0-day, group; 28 (25%) in the 1-day intervention group; 24 (22%) in the 2-day intervention group; and 28 (25%) in the 3-day intervention group. The Kolmogorov–Smirnov test indicated that the study variables were normally distributed across the four groups at baseline (*p* > 0.05), allowing for the use of an ANOVA to assess the baseline characteristics of the patients. This analysis further confirmed that the characteristics were balanced across the four control and experimental groups.

### 3.2. In-Brace Correction Ratios

As shown in Figure 5, for the thoracic scoliosis curves, the mean in-brace correction ratio was significantly improved in the 2-day intervention (0.83 ± 0.76) and 3-day intervention (0.81 ± 0.11) groups as compared with the 0-day control group (0.68 ± 0.13) (*p* < 0.01). Additionally, the mean in-brace correction ratio was significantly different between the 1-day intervention (0.70 ± 0.12) and 3-day intervention (0.81 ± 0.11) groups (*p* < 0.05).

Similarly, for the lumbar and thoracolumbar curves, the in-brace correction ratio was significantly increased in the 2-day intervention (0.85 ± 0.11) and 3-day intervention (0.85 ± 0.13) groups compared with the 0-day control group (0.69 ± 0.19) (*p* < 0.05). No significant differences were observed in other pairwise comparisons (Figure 5).

### 3.3. Patient Comfort Level While Wearing the Fitted Brace

The results of the ANOVAs and LSD post-hoc tests revealed significant differences regarding the patients’ comfort levels while wearing the fitted brace. In particular, the mean comfort score was substantially increased in the 2-day intervention (6.96 ± 1.31, 36.2% increase) and 3-day intervention (7.43 ± 1.48, 45.4% increase) groups as compared with the 0-day control group (5.11 ± 1.10) (*p* < 0.001). Significant improvements in the mean comfort score were also observed in the 2-day intervention (6.96 ± 1.31, 21.0% increase; *p* < 0.01) and 3-day intervention (7.43 ± 1.48, 29.2% increase; *p* < 0.001) groups compared with the 1-day intervention group (5.75 ± 1.40) (Figure 6).

### 3.4. Impact of the Number of Vertebrae Involved in Scoliosis Curvature on the In-Brace Correction’s Effectiveness

The participants with thoracic scoliosis had a median number of seven vertebrae within a scoliotic curvature, while those with lumbar/thoracolumbar curves had a median number of six. Based on these medians, subjects were further divided into different groups: thoracic curves (number of vertebrae <7 and ≥7) and lumbar/thoracolumbar curves (number of vertebrae <6 and ≥6). Due to the lack of significant differences between the control group and 1-day intervention group and between the 2-day and 3-day intervention groups, the analysis was further divided into two broader categories for comparison: 0–1 day of intervention vs. 2–3 days of intervention (Figure 7a).

For the 0–1-day intervention group, patients with thoracic curves involving ≥7 vertebrae had a significantly better in-brace correction ratio than those with <7 vertebrae (*p* = 0.003). Similarly, in the 2–3-day intervention group, patients with ≥7 vertebrae in their thoracic curves also showed a significantly better correction ratio than those with <7 vertebrae (*p* < 0.001). A similar pattern was observed in the lumbar/thoracolumbar curve groups. In both the 0–1-day and 2–3-day intervention groups, patients with ≥6 vertebrae in their lumbar/thoracolumbar curves exhibited significantly better in-brace correction ratios than those with <6 vertebrae (*p* < 0.001) (Figure 7b).

Correlation tests confirmed that, for patients receiving 0–1 day of the intervention, there was a low positive correlation between the number of vertebrae and the initial in-brace correction ratio for both the thoracic (r = 0.248, *p* = 0.004) and the lumbar/thoracolumbar (r = 0.075, *p* = 0.049) curves. For patients receiving 2–3 days of the intervention, there was a low positive correlation between the number of vertebrae and the initial in-brace correction effectiveness for the lumbar/thoracolumbar curves (r = 0.220, *p* < 0.001), whereas no correlation was observed for the thoracic curves (Figure 8).

## 4. Discussion

This study evaluated the therapeutic effects of an innovative 3D spinal decompression and correction device in patients with AIS and explored the impact of the number of vertebrae within the scoliotic curvature on the effectiveness of the in-brace correction by conducting a single-centre, double-blinded RCT on 110 patients with AIS. By preliminarily integrating the traditional brace therapy with more advanced assistive robotic technology, this study delved into a new domain in the conservative treatment of scoliosis, offering a more effective and user-friendly treatment option. Although the current results are preliminary, the current study still provides valuable insights for the future development and optimization of treatment strategies. More details can be found below.

The primary finding of this study is that applying the novel 3D spinal decompression and correction device in patients with AIS could significantly enhance in-brace correction ratios. Notable improvements in the in-brace correction ratios were observed for both the thoracic and the lumbar/thoracolumbar curves. In particular, the treatments lasting for more than 2 consecutive days demonstrated more significant effects than that of no treatment or only 1 day of treatment in participants. This could be explained by increased spinal flexibility [21], which was supported by our previous pilot study that showed that the proposed treatment could significantly increase the flexibility of the trunk in forward bending and lateral bending tests in patients with AIS [22]. A previous systematic review also supported that, among all factors, increased spinal flexibility has been a favourable predictive factor for improved initial in-brace corrections [23]. This review supported our hypothesis and the positive effect of the developed treatment protocol on treating scoliosis. To further verify this finding, future studies may consider utilizing some non-invasive evaluation techniques, such as ultrasound imaging [24,25,26] and photogrammetry [27], to evaluate and compare the difference in scoliotic curvature before and after the treatment. While the thoracic in-brace correction ratio continued to increase after receiving the treatment for more than two days (i.e., 1-day vs. 3-day treatment), there was no significant difference for the lumbar/thoracolumbar curves. This may be because traction and lateral forces have a more direct impact on the soft tissues of the lumbar spine, allowing for quick results with short-term treatments [28]. However, for the thoracic spine, traction appeared to take a longer time to be effective. Additionally, due to the ribs, the effect of lateral forces in releasing the soft tissues of the thoracic spine gradually improved as treatment progressed [29]. Future studies need to further investigate treatment dosage and changes in soft tissues during the treatment process to validate our hypothesis.

The secondary finding of this study is that AIS patients’ comfort level during the initial brace fitting, as measured directly using a VAS, significantly improved after receiving treatment with the novel 3D spinal decompression and correction device. This improvement was especially evident in patients who received more than two days of treatment, which may also be related to the increased spinal flexibility following the treatment. The combination of traction with substantial lateral corrective forces allows for the stretching of the original tense paravertebral muscle contractures, thus reducing the discomfort of affected muscles during the initial brace fitting. Previous studies have reported that discomfort while wearing a brace affected the treatment outcome due to excessive friction and pressure [13] and reduced wearing time [14] in patients with AIS. The improved appearance and wearing comfort level of the brace can significantly improve patients’ psychological acceptance and compliance [30]. A systematic review by Negrini et al. also indicated that brace treatment did not significantly improve the quality of life or mental health status of adolescents [7]. While the VAS is a commonly used tool to assess comfort levels, it is important to note that its results can be influenced by individual differences, such as pain tolerance, mood, and personal perceptions, which may limit its reliability and standardization [20]. Therefore, future studies may need to cross-validate the findings with additional assessment methods to provide a more comprehensive evaluation of treatment effectiveness. The positive findings of the current study supported the feasibility of reducing patients’ discomfort levels during their initial brace fitting by utilizing this novel 3D spinal decompression and correction technology. Significantly improved comfort of patients during the initial brace fitting can help them adapt to the brace treatment more quickly. This improvement is expected to enhance patients’ acceptance of brace treatment, increase compliance, elongate effective brace wearing times, and ultimately, enhance the overall effectiveness of conservative brace treatments for scoliosis. Further studies are still needed to explore and identify an optimal treatment protocol that balances not only treatment outcomes but also cost-effectiveness for patients. Long-term studies investigating the patient compliance in wearing the braces after receiving the developed novel 3D spinal decompression and correction treatment could also be conducted in the future.

It is also interesting to observe that a greater number of vertebrae involved within a scoliotic curvature has been associated with improved effectiveness of in-brace correction in patients with AIS. This is in line with a previous study that evaluated and reported a significant, negative correlation between the coronal deformity angle ratio (C-DAR) and in-brace correction (r = −0.69), indicating that in-brace correction is less effective in patients with a higher C-DAR [31]. The finding of the current study further supported that a reduced number of vertebrae may contribute to a smaller coronal deformity angle ratio, leading to reduced effectiveness of in-brace corrections. This may be due to the fact that fewer vertebrae within one scoliotic curvature may lead to a shorter lever arm and/or reduced space to provide the corrective force or pressure via the brace. This finding suggests that the patient’s anatomical spinal characteristics, particularly the number of vertebrae within one specific scoliosis curvature, need to be carefully revised and considered when developing treatment plans. 

There are several limitations that need to be acknowledged for this pilot study. Firstly, all the braces were designed using a CAD system and simulated within the computer system to ensure that the simulated effects could achieve full correction of the spine. However, there were discrepancies between the computer simulations and the actual initial fitting results. Such differences may be related to the varied muscular states of different patients’ trunks, and it is anticipated that the developed device could potentially reduce these discrepancies to some extent. Secondly, the sample size was relatively small for a clinical trial, and only a single medical centre was involved in data collection. Further validation is still required through larger-scale and multi-centre studies. Additionally, this study lacks a long-term follow-up to monitor the prolonged compliance with brace wearing and long-term changes in the Cobb angle in participating patients with AIS. A muti-centre RCT with a large sample size, long-term follow-up, and high-quality study design will be conducted in the future to identify the optimal treatment for patients with AIS. 

## 5. Conclusions

In summary, the results of this study supported the notion that using the novel 3D spinal decompression and correction device can significantly enhance and in-brace correction’s effectiveness and a patient’s comfort level when wearing the brace in patients with AIS. Additionally, the number of vertebrae was positively correlated with the effectiveness of the in-brace correction during the initial fitting of spinal orthosis for patients with AIS. The findings of this pilot study could provide important clinical guidance for future treatment of scoliosis and offer valuable insights for future studies and clinical practice in this field.

## Figures and Tables

**Figure 1 bioengineering-11-01246-f001:**
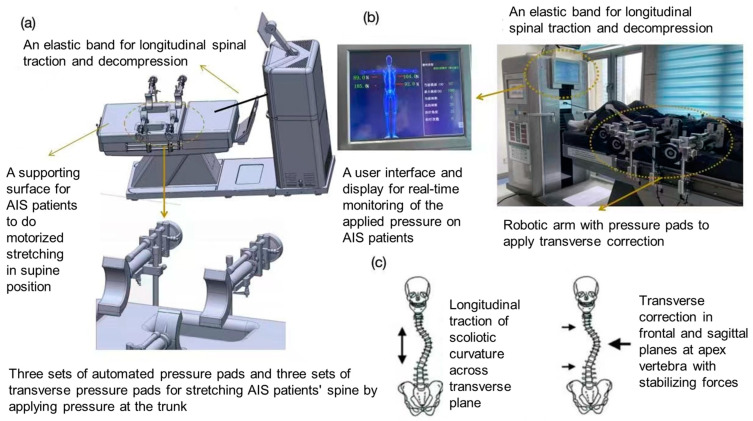
Diagram of the 3D spinal decompression and correction device: (**a**) schematic diagram of the novel 3D spinal decompression and correction device; (**b**) an example of applying automated spinal longitudinal traction, lateral correction, and axial rotation protocol in a patient with AIS; and (**c**) application of longitudinal traction and transverse correction on a scoliotic curve.

**Figure 2 bioengineering-11-01246-f002:**
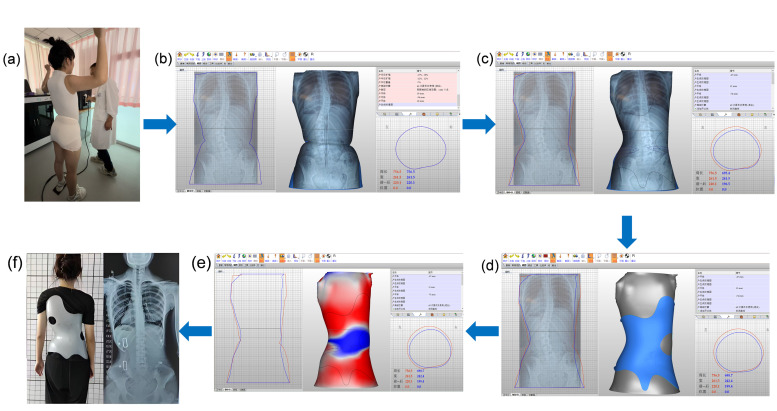
Manufacturing process of the scoliosis brace based on CAD/CAM for each AIS participant: (**a**) scanning of a patient with AIS; (**b**) importing the scanned model and the patient’s X-ray into the software for brace design; (**c**) computer-aided simulation of the correction outcome; (**d**) illustration of the brace’s appearance; (**e**) computer-aided simulation of the pressure applied by the brace on the patient’s body surface; and (**f**) brace fitting and in-brace X-ray of the same patient with AIS. Explanation of lines and regions in (**b**–**e**): the red line represents the original spinal model contour, and the blue line represents the adjusted contour. In (**b**), before design adjustments, the red and blue lines overlap. In (**e**), the red regions indicate areas of pressure applied by the brace, with darker shades representing higher pressure levels, while blue regions represent areas where pressure has been relieved. White regions correspond to areas with no significant pressure adjustment.

**Figure 3 bioengineering-11-01246-f003:**
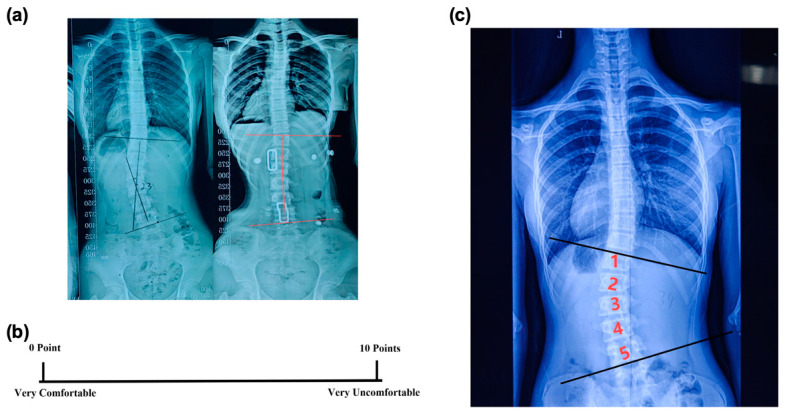
Primary and secondary outcome measures: (**a**) measurement of the pre-brace Cobb angle and in-brace Cobb angle; (**b**) comfort level assessed using a visual analogue scale (VAS); and (**c**) number of vertebrae within a scoliotic curvature.

**Figure 4 bioengineering-11-01246-f004:**
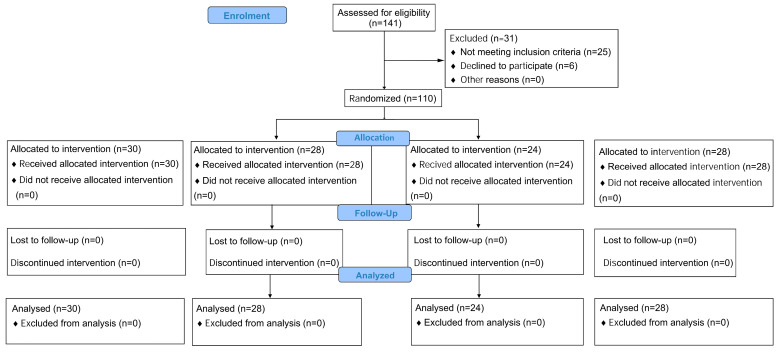
CONSORT 2010 flow diagram of this study.

**Figure 5 bioengineering-11-01246-f005:**
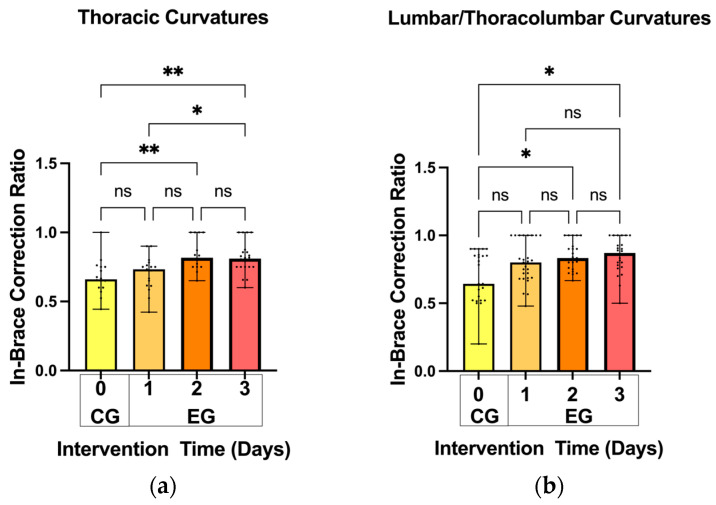
Comparison of the in-brace correction ratios among different control and experimental groups for the (**a**) thoracic and (**b**) lumbar/thoracolumbar curvatures. (Note: ns: no significance; *: *p* < 0.05; **: *p* < 0.01; CG: control group; EG: experimental group).

**Figure 6 bioengineering-11-01246-f006:**
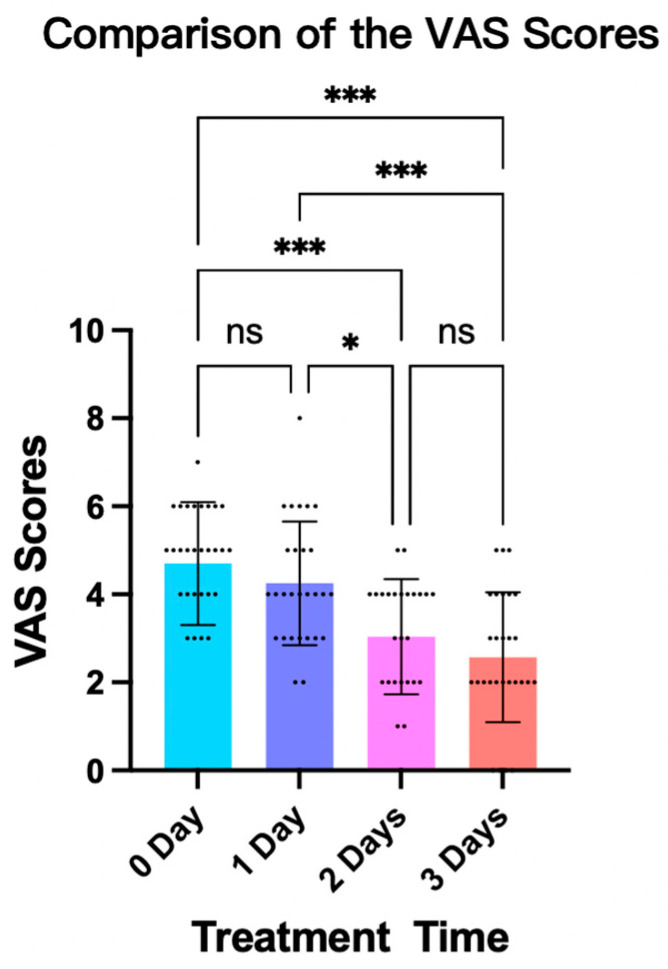
Comparison of the VAS scores of patients’ comfort levels. (Note: ns: no significance; *: *p* < 0.05; ***: *p* < 0.001).

**Figure 7 bioengineering-11-01246-f007:**
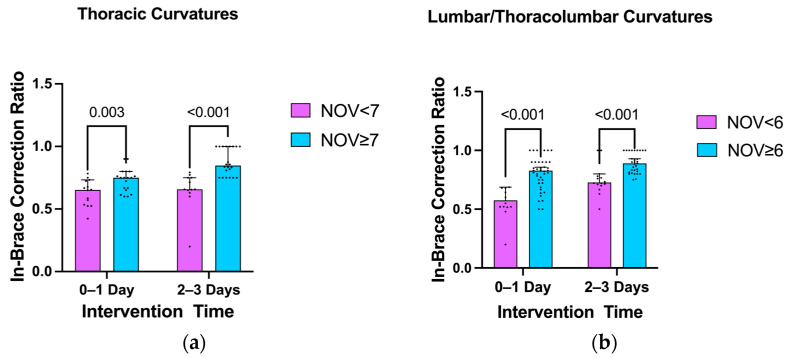
Comparison of the in-brace correction ratio in the (**a**) thoracic and (**b**) lumbar/thoracolumbar curvatures with various numbers of vertebrae (NOV).

**Figure 8 bioengineering-11-01246-f008:**
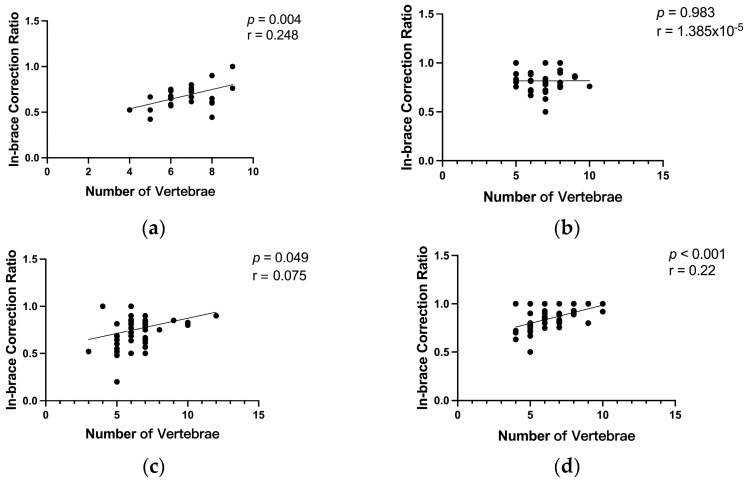
Correlation between the number of vertebrae and the in-brace correction ratio in (**a**) thoracic curves for the 0–1-day intervention group; (**b**) thoracic curves for the 2–3-day intervention group; (**c**) lumbar/thoracolumbar curves for the 0–1-day intervention group; and (**d**) lumbar/thoracolumbar curves for the 2–3-day intervention group.

**Table 1 bioengineering-11-01246-t001:** Baseline characteristics of the patients (*n* = 110).

Subject Demographics	0-Day Control (n = 30)	1-Day Intervention (n = 28)	2-Day Intervention (n = 24)	3-Day Intervention (n = 28)	*p*-Value
					(ANOVA)
Age (years), Mean (SD)	13.97 (2.03)	14.14 (1.60)	14.79 (2.65)	14.71 (1.60)	0.325
Gender, n (%)					
Female	20 (18.20%)	23 (20.90%)	17 (15.50%)	23 (20.90%)	
Male	10 (9.09%)	5 (4.55%)	7 (6.36%)	5 (4.55%)	
Curve type, n (%)					
STC	3 (2.73%)	2 (1.82%)	6 (5.45%)	5 (4.55%)	
SL/TLC	12 (10.91%)	9 (8.18%)	8 (7.27%)	5 (4.55%)	
DC	15 (13.64%)	17 (15.45%)	10 (9.09%)	18 (16.36%)	
Risser Sign, Mean (SD)	2.60 (1.16)	2.71 (1.08)	2.81 (1.17)	2.79 (0.96)	0.940
Cobb angle, Mean (SD)					
T Cobb angle	24.78 (7.22)	25.89 (7.08)	28.58 (9.16)	29.36 (6.89)	0.091
L/TL Cobb angle	25.52 (7.24)	27.27 (7.95)	28.39 (7.18)	30.74 (7.32)	0.104

(SD: standard deviation; T: thoracic curvature; L: lumbar curvature; TL: thoracolumbar curvature.)

## Data Availability

The data that support the findings of this study are available from the corresponding authors upon reasonable request.

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
