# Peer review of "Digitalized 3D Spinal Decompression and Correction Device Improved Initial Brace Corrections and Patients’ Comfort Among Adolescents with Idiopathic Scoliosis: A Single-Centre, Single-Blinded Randomized Controlled Trial"

_bioengineering, 2024, doi:10.3390/bioengineering11121246_

Round 1
Reviewer 1 Report
Comments and Suggestions for Authors
I think the manuscript is overall good and the conclusion is sound and supported by the data. The major concern is the poor image quality provided in figure 2 and figure 3. Please address this issue to improve the quality of the manuscript. A small question is why the number of female patients is more than male patients in each study?
Reviewer 2 Report
Comments and Suggestions for Authors
This work describes a study to evaluate a novel three-dimensional (3D) spinal decom
pression and correction device in improving the in-brace correction and patient comfort level. The article is well written and well organized. I consider that this article is ready to be published after minor revisions:
The title should be shorter and more concise.
How are longitudinal traction and transverse correction loads applied by the device? Are they dynamic loads? What are the amplitude and frequency values? The duration of treatment is 30 minutes, twice a day, but the remainder remains to be described to allow the study to be repeated. How you decide these treatment parameters? Based on other works?
Reviewer 3 Report
Comments and Suggestions for Authors
Review
Bioengineering (ISSN 2306-5354)
Title:
Effectiveness of a Novel Three-Dimensional Spinal Decom-pression and Correction Device in Improving Initial Brace Correction and Patient Comfort among Adolescents with Idiopathic Scoliosis: A Single-Centre Single-Blinded Randomized Controlled Trial
1. What is the main question addressed by the research?
This study aimed to evaluate the efficacy of a novel three-dimensional (3D) spinal decom pression and correction device in improving the in-brace correction and patient comfort level for adolescents with idiopathic scoliosis (AIS), and to assess the impact of the number of vertebrae involved in the scoliotic curve on correction effectiveness
2. What parts do you consider original or relevant for the field? What
specific gap in the field does the paper address?
This study is significant as it presents a single-blinded randomized controlled trial involving 110 patients using a novel three-dimensional (3D) spinal decompression and correction device.
3. What does it add to the subject area compared with other published
material?
This paper provides a detailed description of the process and method of developing the 3D device as well as its application. It is considered a high-quality study based on robust data obtained from sufficiently sized experimental and control groups.
4. What specific improvements should the authors consider regarding the
methodology? What further controls should be considered?
As a single-blinded randomized controlled trial based on adequately sized experimental and control groups, this study demonstrates a satisfactory research design and high-quality statistical data processing, leaving no need for further revisions.
.
5. Please describe how the conclusions are or are not consistent with the
evidence and arguments presented. Please also indicate if all main questions
posed were addressed and by which specific experiments.
The study features a sufficient sample size and an appropriate research design, providing a clear basis for statistical processing. Consequently, the reliability of the resulting data is high.
6. Are the references appropriate?
No errors were observed in the reference section, so no additional revisions are necessary.
7. Please include any additional comments on the tables and figures and
quality of the data.
If the computer screen in Figure 2 is directly downloaded and used, higher-quality images could be provided.
